# From Disruption to Dialog: Days of Judaism on Polish Twitter

**Mariusz Pisarski** [1,*] **and Aleksandra Gralczyk** [2]

1  Department of Media and Journalism, University of Information Technology and Management in Rzeszów, 35-225 Rzeszów, Poland
2  Institute of Media Education and Journalism, Cardinal Stefan Wyszynski University in Warsaw, 01-815 Warszawa, Poland; a.gralczyk@uksw.edu.pl
*  Correspondence: mpisarski@wsiz.edu.pl

**Abstract:** While social media platforms afford visibility to marginalized voices and enable dissemination of alternative narratives, their own "power laws" can make few users responsible for most of the attention. New power users can redirect discussion away from those who initiate a conversation. The aim of this study is to examine relations between the network "gatekeepers" and "gatewatchers" following the announcement of the Days of Judaism celebrated by the Polish Episcopate every January. Two methodological approaches were taken over two consecutive years: social network analysis (SNA), and linguistic analysis of social media discourse. The linguistic analysis confirmed importance of classical rhetoric effects on Twitter. The social network analysis revealed that a balanced, personal statement given by users with high network standing outside of the Twittersphere can ignite constructive dialogue in the spirit of the inter-religious exchange that the idea behind Days of Judaism stands for. Our conclusion is that a careful social media policy of the Church, a controlled engagement in the public conversation, possibly by lay sympathizers of high standing in the real public life, have the potential for dispensing with the infamous toxicity of Twitter, and for turning conversation on any topic, even the most controversial, into positive exchange within the community of believers.

**Keywords:** discourse studies; social network analysis; inter-religious dialog; new media studies; toxicity on social networks; Judaism; anti-Semitism

## 1. Introduction

### 1.1. Background

Twitter and other social media platforms break away from the traditional separation between producers and consumers to transform their users into partisan broadcasters. As such, Twitter is considered a space for cultural conversation (Brock 2012, p. 541). With real-time updates and comments being its focal point, it is not surprising that the social media service has become a popular barometer by which media researchers, sociologists and journalists measure collective sentiment of the "Twittersphere" in regard to any chosen topic.

The point of departure of our study is that Twitter and other social media platforms are not only networking tools, but—quite prominently—language tools. This premise requires us to combine two methodological approaches. One is indebted to social network analysis (SNA), which focuses on relations between users and their accumulated ranking in the discursive field, as well as in detecting the ad hoc formation of online communities used in graph and network theory. The second approach borrows from linguistic studies of new media and its rhetorical affordances and constraints in order to apply classical notions of rhetorical effects of *ethos*, *pathos*, and *logos* within the social media context (Berlanga et al. 2013, p. 133). Both approaches are viewed in this paper in a broader, historical context of public conversation of the Church in modern media, including traditional media predating the Internet.

Few key findings about the structure and logic of Twitter discourse are relevant to our research. Firstly, one needs to acknowledge that social media platforms afford visibility to marginalized voices and enable alternative narratives of dissent (Hamdy and Gomaa 2012, pp. 196–7). This structural affordance—defined as cultural use of the medium and what is possible within it (Norman 1999, p. 39)—came into prominence during the Arab Spring uprisings of 2011 and has led to the formulation of network gatekeeping and network gatewatching theories. Networked gatekeeping is defined as "multiple levels of relationships and symmetries between variant news actors who hold diverse levels of power and positions" (Barzilai-Nahon 2008, p. 1494). Gatewatching, on the other hand, suggests a novel way of curating media content by active audience members, which provides further filtering and amplifying of news items (Bruns 2005, p. 2).

Secondly, existing research unveiled an important power law in networks (Barabási and Albert 1999; Perline 2005) defined as a situation where few users are responsible for the majority of attention (Singh and Jain 2010; Meraz and Papacharissi 2013). Such power law leads to formation of elite users within a horizontal, dispersed, and multi-directional realm of media communication. Elite users become alternative actors positioning themselves between traditional media outlets and the audience. By accumulating groups of followers, they are able to create new, virtual communities. This phenomenon, visible in our examples in the next section, directly relates to yet another important finding about the nature of networked communication, that of homophily, the tendency to follow like-minded individuals prevalent on Twitter. Replies and mentions between like-minded users strengthen a group's identity and political affiliation (Yardi and Boyd 2010, p. 318).

Thirdly, it is important to distinguish between the static and dynamic, or accumulative and emergent, aspects of social media. On the one hand, both the presence of official, institutional news outlets and the formation of elite social media users point to the importance of accumulated merit for one's standing within the network. Numbers of followers and likes constitute static, cumulative data, which promotes visibility of a given social media account over time and—potentially—in any networked conversation that follows. On the other hand, many scholars point to the prevalence of the second major force shaping the network discourse—the formation of ad hoc publics (Bruns and Burgess 2011). Any event and any topic create their own audience with a fresh configuration of actors. Skillful use of Twitter's stylistic and rhetorical affordances can—theoretically—elevate even a random user who joined the discussion by chance to a prominent position within the network. Ad hoc clustering of group identities points to temporal, localized, and tactical aspects of homophily, which from this perspective looks as something not given and stable but volatile. This means, firstly, that on Twitter "a rhetor is only as good as his most recent tweets" (Swift 2010, p. 19), but secondly, that persistent engagement with one's audience can rewrite the logic of network "noise" and secure even traditionally inclined media outlet, such as EpiskopatNews we look at in this paper, to a status of a strong player within the Twitterverse.

Networked publics are typically called into being on Twitter using text, hashtags, and addressivity markers, which shape the flow of information stream. Studying these markers, which on Twitter take the form of mentions and retweets marked by "@", "RT", and "via" labels, can reveal media landscapes quite different from the ones we are used to. It is populated by emerging public commentaries and opinions, including those traditionally kept in shadow by the power structure of official news sources. Exposing main actors and their strategies and the communities of like-minded individuals they attract can reveal not only diverse, perplexing, politically incorrect ideas among Catholic congregations in Poland but also point toward some new forms of engagement, a dialogue, a possibility of reconciliation between diverse voices of the same networked environment of "Catholic" Twitter and beyond.

*1.2. The Evolution of the Church's Communication with the Faithful through the Media*

Since its inception, the Church has tried to evangelize using all possible means of interpersonal communication. However, prior to the appearance of media, contact with its members was greatly limited. Initially, it consisted of oral communication, life testimonies, letters, sacramental and liturgical rituals, religious symbols, and songs, as well as church buildings and their interiors. In terms of dialogue in any form with the faithful, the breakthrough moment was the invention of printing by Johannes Gutenberg in 1455. However, both the church and political authorities very quickly found it to constitute a threat, as it allowed reproduction of content they found inconvenient. Therefore, both the secular and church authorities sought control over the duplication of periodicals and books. They introduced a system of censorship, which resulted in a lack of space and opportunities for dialogue with the faithful. Simultaneously, as access to various books improved at the end of the fifteenth century, the Church lost its leadership role in disseminating and sharing knowledge, which, in turn, had an impact on the split of Christianity (De Vaujany 2006, p. 359).

The Church's position on mass media, which started with the appearance of the large circulation press in 1814, was unfavorable from the very beginning. Its first official statement of position came with Pope Gregory XVI's publication of the "Miriari vos" encyclical in 1832, which condemned the dissemination of doctrines going against the Church's teachings and the promotion of its separation from the state (Lewek 2003, p. 47). However, attitudes towards the press evolved over the years. In 1850, the first issue of La Civilta Cattolica was published in Rome (Dante Francesco 1990, p. 57). The Church had a similar initially reluctant attitude towards the medium of film, which can be dated back to 1895. The Church saw film as a demoralizing tool used for spreading indecency and warned the faithful of its demoralizing influence. However, its approach changed over the years. Already in 1928, Pius XI wrote in the encyclical Divini Illius Magistri that the Church supports the media provided that they are conducive to Christian education. Furthermore, in 1938, the Holy See published Pius XI's encyclical devoted entirely to film, which featured a slightly less radical attitude towards this medium. Therein, the Pope encouraged the production and distribution of films that could serve evangelization (Lewek 2003, p. 68).

As the above shows, in the first period of mass media's development, the initial phase consisted of attempts to protect the Church and fight with the media. This attitude stemmed from a deep fear and anxiety towards new inventions. For a long time, Church hierarchs failed to accept that the faithful could obtain information and knowledge other than content inspired by Christian science and morality, or to see that radio and film do not always breed evil. Dialogue of church members with the Church through the press and film still remained impossible.

The appearance of two more forms of mass media—radio in 1869 and television in 1927—led to a shift in the Church's attitudes towards media. Practically from the beginning, the Church spoke favorably of these two new tools, which were seen to not only have great potential in apostolic work, but also as a space for dialogue with the faithful. For Pope Pius XII, radio was a tool for uniting people, as it could reach millions and immediately, simultaneously garnering their interest in the common cause (Pokorna-Ignatowicz 2002, p. 45). In 1931, Vatican Radio started (De Vaujany 2006, p. 359), and over the following years, the Church created its own Catholic stations all around the globe. It has been continuously broadcasting ever since, also on secular radio stations, usually at fixed times. The broadcasts usually come in the form of Holy Masses, church ceremonies, and reports from papal pilgrimages, as well as opinion pieces, reports, and comments, and they are not strictly religious, but mostly show content related to social and political life, as well as culture and the arts, all presented in an interesting, unobtrusive form.

In turn, the first religious television program was the broadcast of the Midnight Mass from Notre Dame in Paris in 1948. Two years later, in 1950, the French government gave the Holy See a television station that broadcast Jubilee celebrations and related events (Pokorna-Ignatowicz 2002, p. 47). In the initial stage of television's development, the

Church did not wish to establish its own stations, but instead sought to include religious programming on secular ones. Most often, these were broadcasts of religious celebrations, papal pilgrimages, sermons, or discussions of social issues.

When radio and television were developing, the Church's approach to media began to evolve. The attitude changed, from total negation and distrust to one of openness to the possibility of communicating with the faithful, a stage of dialogue. Lay people, followers of various religions, members of different cultures and communities began to react to the Church's voices and positions. The faithful gained a voice in the press or radio, but their message was still subjected to a moral assessment by the Church.

Over the following years, the Church warmed even more to communicating with church members through the media. *The Pastoral Instruction Communio et Progress*io of 1971 and the *Instruction Aetatis nove* published in 1992 were significant documents, in many respects valid to this day and changing the Holy See's approach. "The media facilitate a broad dialogue within the church, shape the public opinion within the church and enable the Church to conduct dialogue with the world, give the world an insight into the life and activities of the Church, while also informing the Church about the world and the signs of the times" (Pastoral Instruction Communio et Progressio, Rome 1971).

A turning point for the faithful who wanted to participate in the Church's activities on an ongoing basis was the Holy See's great enthusiasm in accepting the Internet. As was the case with radio and television, the Church saw the Internet's great potential mainly in its guaranteed immediate reach of millions. The worldwide network also allows thoughts and views to be exchanged and serves as a platform for dialogue and shaping public opinion. In 2001, John Paul II was the first pope to use e-mail when he sent out the apostolic exhortation Ecclesia in Oceania (Raś 2021). The following years saw intense development of mobile technologies. The media ceased to be something one simply consumed, as new media allow the recipient to create, to be part of the media world, an element of communication through dialogue. In January 2009, the Holy See launched its official YouTube channel. However, it did not resign control over the tools. The faithful cannot comment on content posted on the Holy See's websites or on its YouTube channel (Cheong and Ess 2002, p. 15).

Social media provide the greatest opportunities for expressing oneself and commenting freely. The first Pope to post on Twitter was Benedict XVI, whose first tweet as @Pontifex appeared on 12 December 2012. Today, Pope Francis continues such activities. This channel allows almost immediate contact with the faithful, informing them of important events, and sharing opinions. Social media are often a place for church members to meet with a blogging bishop or priest, which means the Church hierarchs are becoming partners for discussion, not only creators and senders of messages as before. Furthermore, religious practices are not limited by place and time. A telephone with permanent access to the Internet is enough to participate in religious events or comment on them. The Day of Judaism, which the Polish Church has been celebrating every year since 1998 on 17 January, on the eve of the Week of Prayer for Christian Unity, is one such event. The first service under the Day of Judaism held by the Catholic Church was in Cracow; it was a word of God service, and only 50 people were in attendance. However, as the years passed, the celebration has gained popularity among the faithful and the media. Services are prepared, and everyone can participate, as live broadcasts via Facebook or YouTube pages of the individual archdioceses are available. All the largest national and local stations prepare reports from these days. Recipients can share their thoughts, opinions, and views, as well as take part in a number of accompanying events, most often publicized by the local media (with each archdiocese preparing its own), such as symposia, workshops for history and religion teachers, film screenings, lessons in museums, or music evenings. The faithful are also informed of the celebrations via newspapers, in which they can read announcements or reports from the celebrations, or through social media, which give them the opportunity to immediately comment.

### 1.3. The Nature of Interreligious Dialogue

The teachings of the Second Vatican Council were the start of a transformation in the relations between the Catholic Church and other religions. This approach influenced the initiation of interreligious dialogue based on a desire for mutual understanding and respect. Such a perspective was reflected in the Council's Declaration on the Church's Relationship to Non-Christian Religions, "Nostra aetate", in the Declaration on Religious Freedom, "Dignitatis humanae", and in the Dogmatic Constitution on the Church, "Lumen gentium" (Cassidy 2001, p. 11). "Thus the Church of Christ acknowledges that, according to God's saving design, the beginnings of her faith and her election are found already among the Patriarchs, Moses and the prophets. She professes that all who believe in Christ—Abraham's sons according to faith (6)—are included in the same Patriarch's call, and likewise that the salvation of the Church is mysteriously foreshadowed by the chosen people's exodus from the land of bondage. The Church, therefore, cannot forget that she received the revelation of the Old Testament through the people with whom God in His inexpressible mercy concluded the Ancient Covenant. Nor can she forget that she draws sustenance from the root of that well-cultivated olive tree onto which have been grafted the wild shoots, the Gentiles. Indeed, the Church believes that by His cross Christ, Our Peace, reconciled Jews and Gentiles. making both one in Himself" (Nostra aetate 1965, p. 4).

From that moment on, the Catholic Church tried to implement the post-conciliar ideas contained in the above-mentioned documents. The activities became more intense and deeper during the pontificate of John Paul II, who believed that contemporary dialogue between religions is not a dialogue between the past (Judaism) and the present (Christianity). This stems from the fact that it is a dialogue held between two living religions within the Church, a dialogue between the first and second part of the Bible (Bartoszewski 2007, p. 15).

Interreligious dialogue can take various forms:

- Dialogue of everyday life—Christians share their joys, sorrows, problems, and worries with people of other religions in the spirit of friendship and openness;
- Dialogue of common endeavor—occurs when Christians, based on their religious needs, cooperate with the followers of other religions on social, economic, and political grounds to protect human rights;
- Dialogue of exchange of religious experiences—the faithful share the spiritual wealth and their own experience of God drawing on personal religious traditions;
- Theological dialogue—the faithful deepen their understanding of their own and other religions, appreciating spiritual and religious values in the spirit of humility and mutual understanding (Gądecki 2002, pp. 18–21).

The objective of interreligious dialogue within the horizontal dimension is to build solidarity on a global scale, while in the vertical dimension, it is a call to unite Christians, so that they bear witness to Christianity's human and religious values (Gądecki 2002, pp. 21–22). "Dialogue cannot take place merely on a horizontal level, being restricted to meetings, exchanges of points of view or even the sharing of gifts proper to each Community. It has also a primarily vertical thrust, directed towards the One who, as the Redeemer of the world and the Lord of history, is himself our Reconciliation. This vertical aspect of dialogue lies in our acknowledgment, jointly and to each other, that we are men and women who have sinned" (John Paul II 1995, p. 35).

### 1.4. Days of Judaism: History and Principles

The creation of the Subcommittee of the Polish Episcopate for Dialogue with Judaism in 1986 (which in 1987 changed its name to a Committee), chaired by Bishop Henryk Muszyński, is seen as the beginning of the modern history of dialogue of the Catholic Church with Jews and Judaism. This happened within the context of a growing conflict over the convent of the Carmelite nuns in Oświęcim (Chrostowski 1999) and one of the key visits in the history of the dialogue, the one of John Paul II to the Greater Synagogue in Rome. "The Jewish religion is not 'extrinsic' to us, but in a certain way is 'intrinsic' to our

own religion. With Judaism therefore we have a relationship which we do not have with any other religion. You are our dearly beloved brothers, and, in a certain way, it could be said that you are our elder brothers" (John Paul II 1986, p. 4).

In 1990, the Christian–Jewish Dialogue Section was established as part of the Polish–Israeli Friendship Society, which has currently been transformed into the Polish Council of Christians and Jews. In 1994, Bishop Stanisław Gądecki (then the auxiliary bishop of Gniezno) assumed the position of the chairman of the Episcopal Commission for Dialogue with Judaism and in 1996 the Council for Religious Dialogue, and within this the Committee for Dialogue with Judaism (Stranz 2007, p. 23). In 1997, during the 291st Plenary Assembly of the Polish Episcopal Conference, the project of the Day of Judaism in the Catholic Church of Poland was presented and approved. Thanks to this, Poland became the second European country after Italy where its celebrations began. Over the years, the initiative was joined by other countries, including Austria, the Netherlands, and Switzerland. The first Day of Judaism in Poland was held on 17 January 1998 (it is a set day for this event). The choice of this particular day is not accidental, as it is the Eve of Prayer for Christian Unity. Such closeness of these events aims to indicate the course of salvation time and the natural close relationship between religions arising from the same roots in Abraham. In addition, it indicates the relationship between the interreligious dialogue in the strict sense and ecumenism. Therefore, dialogue appears as an introduction to ecumenism (Gądecki 2002, p. 134).

Celebrations of the Day of Judaism are intended not only to contribute to the fair and objective presentation of Jews and Judaism, but also to serve as an impulse for spiritual unity between the faithful of both these religions. They are intended to build awareness of common values consisting of faith in one God, trusting the Word of God, universal call to holiness and the tradition of common prayer—both individual and as a community (Gądecki 2002, p. 133). During the first celebration of the Day of Judaism, in a speech at the Nożyk Synagogue in Warsaw, Archbishop Stanisław Gądecki indicated how the faithful of both religions can celebrate this day. The said activities are based on forms of interreligious dialogue and include common prayer, organization of meetings between Jews and Christians allowing them to learn respect and friendship towards the faithful of the other religion, exchanges of their own religious experience on the meaning of life and death, as well as the ways of seeking God. The archbishop also pointed out the possibility of cooperation in various fields aimed at defending human rights, human dignity, justice, and freedom, as well as the opportunity to expand the knowledge of one's own religions on anthropological, philosophical, and theological grounds (Gądecki 1998).

Currently, the Committee for Dialogue with Judaism at the Polish Episcopal Conference, having seen deviations from the original goals, appeals to present and future organizers of these days to observe the following rules:

- Explain and disseminate the nature of the Day of Judaism;
- Bring closer Church's teachings on Jews and their religion after the Second Vatican Council;
- Make prayer an integral part of the Day of Judaism;
- Promote post-conciliar explanations of Scripture, which may have been interpreted in an anti-Judaist and anti-Semitic manner in the past;
- Explain the tragedy of the extermination of Jews to the faithful;
- Show anti-Semitism as a sin (John Paul II);
- Invite representatives of other Churches and Christian communities to common prayer on that day;
- Invite Jews to participate in Day of Judaism celebrations (Committee for Dialogue with Judaism at the 2008 Polish Episcopal Conference 2000).

## 2. Materials and Methods

### 2.1. Visualizing Networks and User Rankings

Data scientists and online researchers point out that it is impossible to maintain neutrality in data visualization (Ben-Murdoch 2013). The same data can be represented in different ways to create different messages, all of which seem "ostensibly trustworthy" (Kennedy and Allen 2016, p. 310). Such a phenomenon is strikingly visible when a network of Twitter users discussing the announcement of the Days of Judaism is mapped in popular visualization tools.

One of such tools, TAGS, offers 3 main views of the network's visual pane, or map. The Default View, based on replies to the original tweet, draws a circular cloud of nodes and connections, namely users and their reactions to each other's tweets. The account initiating the conversation, EpiskopatNews[1], is at the center, surrounded by those who commented, replied, or forwarded the Church's message. Already in the default view user ranking metrics is introduced. The size of each node's label correlates with the Twitter ranking of a user represented by the node; accounts with more followers and more tweets have higher ranking and gain higher visual prominence. The second available layer is the Mentions View. It filters the map of connections by the number of mentions, or appearances, and users' scores in tweets of others. A random tweet with mentions looks like the following:

> @EpiskopatNews @CentrumDialogu @MiastoLodz @YouTube Zastanawiam się kiedy u przedstawicieli Judaizmu obchodzony jest dzień chrześcijan?
>
> [ENG: @EpiskopatNews @CentrumDialogu @MiastoLodz @YouTube I wonder when representatives of Judaism will celebrate the Days of Christianity]

The Mentions View complements the static hierarchy established by the cumulative popularity score of users by foregrounding significant dynamic elements—the number of mentions during a single conversation. Effectively, mentions turn the graph into a popularity vote exclusive to the current discussion. In addition, the visualization algorithm makes the tweets with mentions gravitate towards the user they refer to and—in this way—a cluster of users is formed. Finally, Retweets View of TAGS demonstrates the popularity of a single tweet. This metric refines the clustering of associated nodes even further by adding additional edges (links) between these associations. The network map looks strikingly different from the default, most traditional view. After the filters of mentions and retweets are applied, the network map looks much different than in the initial Default View. Connections take precedence over status, and ad hoc visibility takes precedence over accumulated merit. A general rule of the power law in social media networks is confirmed; only a few users are responsible for most of the attention (Barabási and Albert 1999; Perline 2005). As most users join the discussion in the form a retweet of an existing tweet, their visibility is mostly a function of connection to the author of the quoted tweet. They are represented as small grey dots which support and sustain the network position of other, more vocal, and thus more prominent actors. Meraz and Papacharissi justly call them power users (Meraz and Papacharissi 2013, p. 143).

The node representing EpiskopatNews, which in the first year of our study scored 24 connections (3 tweets, 7 replies, and 17 mentions), was pushed to the background, whereas users with more mentions and replies were taken to the foreground. Other Twitter accounts came to prominence and are above Episkopat news due to their overall "connections" score, ranging from 128 to 35. In this way, the center of Twitter's attention for topics related to Days of Judaism shifted from the official Church sources to alternative views on inter-religious dialogue. One account represents an online opinion portal "Legion, św. Expedyta" with a visible right-wing and nationalist agenda ("God, Honor, Motherland" is its frequent ideological banner). The second of the high scoring nodes in the network represents an influential retired priest known for his radical views on social media. The third position is occupied by an ultra-Catholic right-wing news portal, and the fourth—with a score of 35 connections—is an individual account of a Twitter user of nationalist ultra-patriotic views. All four have moderately high Twitter rankings, with the number of followers at

3000, 8000, 900, and 5000, respectively, compared to 23,000 followers of EpiskopatNews! Of course, social media activity on Twitter is not limited to those most vocal users with a negative attitude towards official Church announcements. If this was the case, our study would inevitably turn into a study of contemporary expressions of Polish anti-Semitism. The network graphs also demonstrate a stable presence of supporters of the Church's message. Apart from affiliated Catholic Church accounts there is also a visibility of lay Catholic groups and individuals, such us independent Catholic media (alatheia.pl), and representatives of local governments (Hanna Zdanowska, president of Łódź), which introduce a crucial balance. Although these accounts might not produce any tweets, they are often mentioned in polemic discussions due to their off-line, public activity supporting the inter-religious dialog, especially if the account has a high Twitter ranking.

For the sake of brevity and clarity, let us label the four users who hijack the conversation with acronyms U1, U2, U3, and U4 (Figure 1). It is worth focusing on the linguistic content of their tweets to look for possible rhetorical reasons for higher attention that Twitter users invested into these messages. The tweet of U2 contains the following:

> *Z uporem godniejszym lepszej sprawy cześć wpływowych hierarchów w Kościele Katolickim lansuje tzw. dzień judaizmu. Ta nierozumna praktyka trwa już 23 lata...* (https://twitter.com/CzarnaLimuzyna/status/1215983702787928064, accessed on 30 January 2020)

> [ENG: *With the stubbornness worth a better case, part of influential hierarchs in the Catholic Church promote so called day of judaism. This unreasonable practice goes back for 23 years . . .* ]

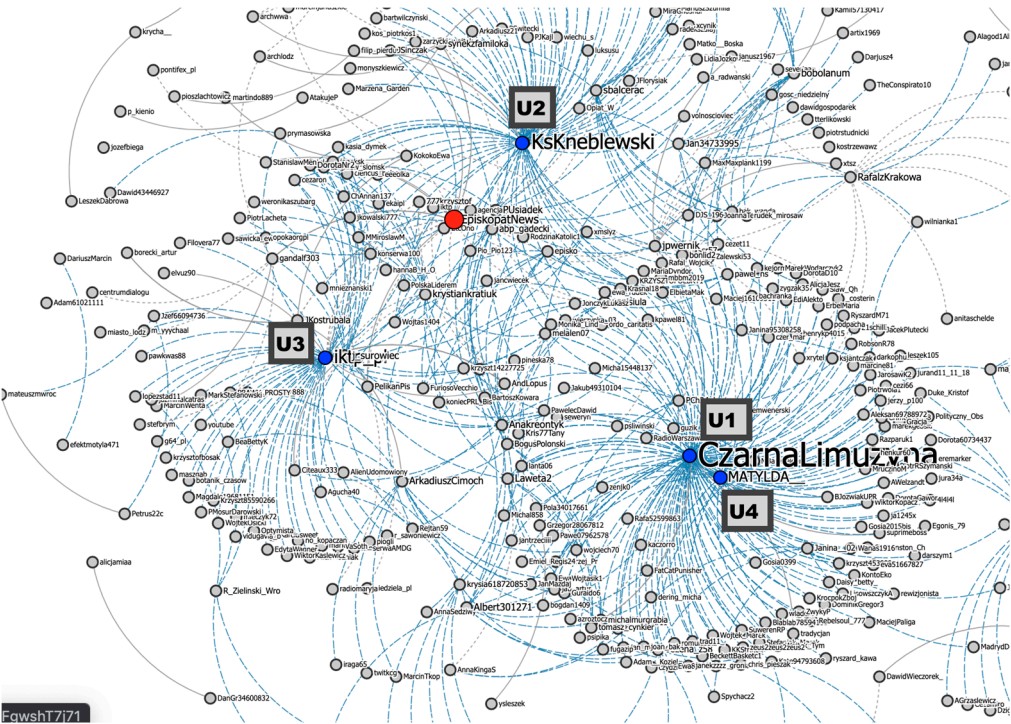

**Figure 1.** The combined Mentions View and Retweet View filters of TAGS. The Polish Episcopate initial post (marked in red) is overshadowed by Twitter noise generated by vocal few users (marked in blue).

With a sophisticated yet catchy exposition of the main argument ("With the stubbornness worth a better case"), a hint of objectivity ("part of church hierarchs"), and an added historical bit of information ("goes back 23 years") the tweet bears marks of professional journalism and objective reporting that targets both the existing audience of the portal and any social media user whose attention it can attract. A suspension at the end of the tweet

introduced by the ellipsis further reinforces the rhetorical strength of the tweet by hinting at a next step, a continuation of the discussion, actions to potentially change the current situation.

U1′s tweet aspires to sound like legitimate commentary from an online news outlet on even footing with any other source of information, EpiskopatNews included. One might argue that this goal is achieved. The tweet includes all three rhetorical components of a successful speech as understood in classical rhetoric and applied to online environments (Berlanga et al. 2013, p. 133): *ethos* in the introductory sentence where cultural standing and objectivity of the author is shown, *logos* in drawing from historical records, and *pathos* in a call to action marked by the ellipsis.

Not far from U1, in terms of both the network ranking and the art of rhetoric, is U2:

*Dni judaizmu, islamu, protestantyzmu. A kiedy „dzień Tradycji" w Kościele?* (https://twitter.com/KsKneblewski/status/1218270854821359616, accessed on 30 January 2020)

[ENG *Days of Judaism, Islam, Protestantism? When to expect "the day of tradition" in Church?*]

A skillful construction of the first sentence sporting a diaphorical type of repetition of different Faiths, the (suggested) rhetorical question ending the second sentence, and the brevity of the whole tweet—make the message highly suitable material for retweeting, ready-made opinion fit for fast redistribution over the network. In this regard, the tweet of U3 is a combination of the strategies employed by U1 and U2:

*Dzień Judaizmu, Dzień Islamu ... A gdyby tak sięgać po temat znacznie bardziej egzotyczny dla naszego duchowieństwa i urządzić w Kościele Dzień Tradycji Katolickiej? Biskupi z @EpiskopatNews celebrujący pontyfikalnie w swych katedrach po staremu, prelekcje ... Niemożliwe, co?* (https://twitter.com/iktp_pl/status/121770987975593 1648, accessed on 30 January 2020)

[ENG: *Days of Judaism, Days of Islam . . . And what if we take a subject much more exotic for our clergy and announce a Day of Catholic Tradition in the Church?* 🫤 *Bishops from @EpiskopatNews pontifically celebrating in the old ways, lectures . . . Not possible, is it?*]

Lack of focus sported by U1 and lack of headline style brevity represented by U2 makes U3 somehow less successful in engaging social media, with only 42 mentions compared to 123 scored by U1. Still, the tweet attracted a visible group of affiliated accounts. The list of "network noise" generated by competing views on the Days of Judaism ends with U4—an account responsible for the first plainly anti-Semitic tweet that barely deserves quoting and surprisingly was not removed by Twitter moderators:

*RZYDZI[2] [sic!] ID NA CAŁOŚĆ. Dzień judaizmu: dzień bez Jezusa i Maryi w Kościele Katolickim. Czas zakończyć ten absurd.* (https://twitter.com/MATYLDA__/status/1216099928537227265, accessed on 30 January 2020)

[ENG: *JEWS GO FULL STEAM AHEAD. Day of judaism: day without Jesus and Mary in Catholic Church. Time to end this absurdity*]

In contrast to U1 and U2 whose tweets were framing the commentary into a context of competing visions of Catholicism in Poland (progressive versus conservative) while still keeping it at an acceptable level of public debate, the tweet published by U4 prefers to target emotions and prejudice rather than reason. Emotive statement at the beginning, calling out representatives of a single specific race as agents of some unspecified doings, then revealing the source for concern (Days of Judaism) and evoking key figures of the Catholic faith as absent (erased? expelled?) due to these doings point to populist rhetoric. The name attached to the account is borrowed from one of the most historically prominent Noblemen families in Poland (Radziwiłł), the account belongs to an attractive young woman, the account's banner includes fragments of a poem by Cyprian Kamil Norwid, one of the greatest Polish romantic poets. Taking into consideration how much of deliberate

tactics of engagement were involved in creation of the—most likely—fictitious account, not only is it comforting that U4 performed quite poorly in terms of network visibility, but it is consistent with previous research that users tend to retweet messages with content rather than with rhetoric. Morally loaded or highly emotive tweets do not get mentioned and retweeted often (Sagi and Dehghani 2014, p. 1351). With 35 retweets and zero replies of U4 it did not manage to attract followers around itself; instead, it is mapped as a radical wing of the sub-network of U1 (Figure 2).

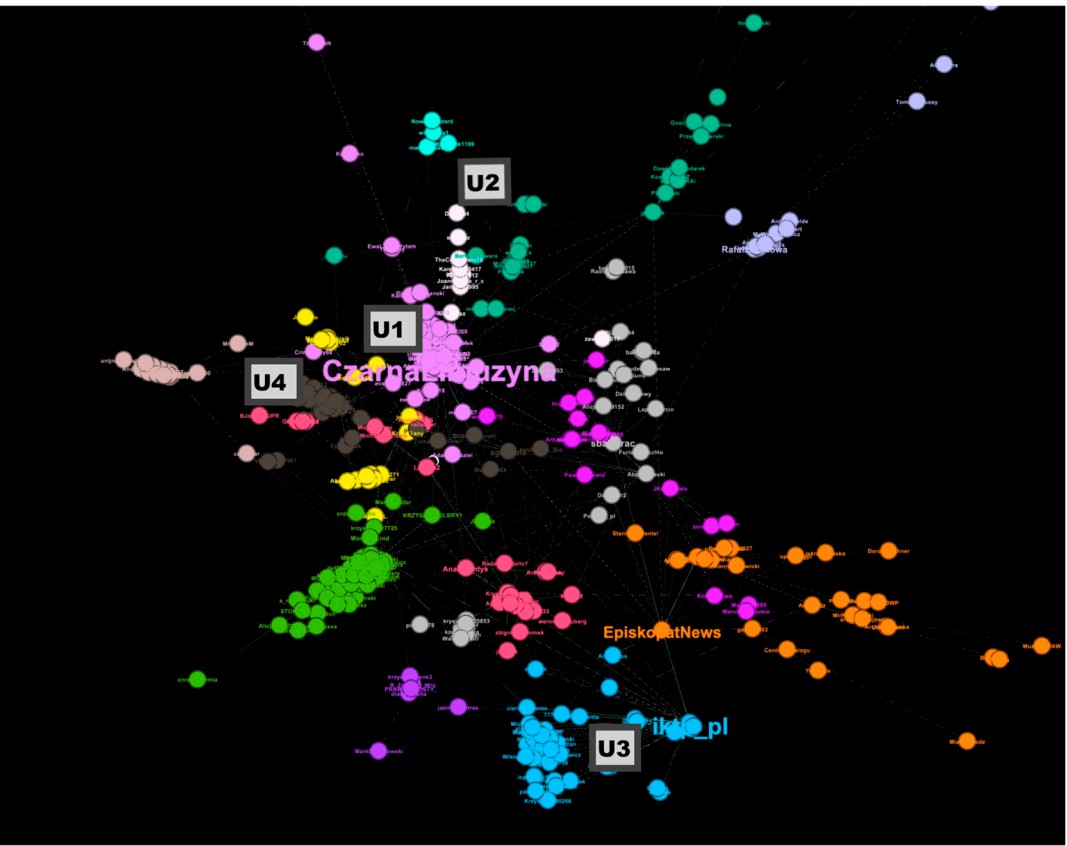

**Figure 2.** Violet, green, and blue subnetworks represent clusters from which the tweets of U1, U2, and U4 arrive, with Episkopatnews (in orange) and affiliate catholic communities (in light purple) pushed towards the right edge of the screen. Closeness of U3 to EpiskopatNews reveals their ideological affiliation despite the former's critical approach to Days of Judaism.

### 2.2. Detecting Communities

The one-to-many model of communication implies verticality of information direction. Many-to-many, on the other hand, supports horizontal conversational practices and the spread of signals across the network (Meraz and Papacharissi 2013, pp. 14–141). It is thus a paradox that network visualization reinstates traditional verticality by prioritizing those actors who—by skillful use of network affordances—achieve the best network scores, in the case of Twitter represented by high enough numbers of mentions, likes, and retweets. The TAGS visualization tool helped us demonstrate that few users can effectively redirect attention and bring exposure to a group of users with certain views at the cost of other groups. Fortunately, the same rule dictates that the damaging effects of a few can also be undone by a few. As a result, a form of second order verticality can be reinstated. One of the network measurements suited for such a task is modularity. This network parameter is used to detect communities active in each network (Lim and Datta 2012, p. 317). For the purpose of our analysis, we used the popular program Gephi through which we analyzed data on mentions and retweets collected by TAGS.

A Python script was employed to convert raw Twitter data gathered by Tags into subsets of data ready to be processed by Gephi. The tool of choice for network studies in Digital Humanities, Gephi—just as TAGS—visualizes such networks parameters as degree and centrality.

However, in contrast to TAGS, it excels in detecting communities within the studied group by visualizing community detection algorithms of degree centrality and modularity. Degree centrality detects the position of a node within the network and its distance to other networks. Modularity measures the strength of connections within the network and consequently partitions it into modules and clusters (Hogan 2016, p. 254). This process clarifies network noise from network data. Extreme voices of lone wolves such as U4 are pushed to the outer perimeter of the graph. In the center, 6 to 8 visible groupings of nodes arise. They correspond to the detected clusters of users with connections to each other formed by retweets and mentions. Mechanisms of retweets and—especially—mentions can uncover communities hidden behind the most vocal users. Users who function as vital nodes in their respective networks, but who did not tweet about the announcement, or whose tweets were not retweeted, are revealed and a new discourse map emerges. Apart from vocal neo-conservatist, nationalistic, right wing, and atheist communities who are responsible for the majority of Twitter noise by re-tweeting catchy, rhetorical messages that question the need for Days of Judaism, few important communities are added to the picture. These are Twitter users who represent the Polish Jewish community, lay Catholic communities, liberal and conservative local politicians and activists, independent Catholic media, and independent lay journalists.

In the data gathered in 2020, communities who support the Church message of inter-religious dialogue and a vision of a multi-cultural Poland were not visibly engaged in conversation. Its tone was set by prominently exposed tweets we have mentioned. A different picture emerged after collecting the data in 2021. In the Twitter conversation surrounding the Days of Judaism of 2021—held in the midst of the second wave of the COVID pandemic—the vocal partisan anti-clericalists of social media were overshadowed by a lively discussion held between liberal and conservative lay Catholics of prominent social standing. The discussion was started, once again, by a skillfully crafted and rhetorically conscious message. Yet this time it came from an avid supporter of inter-religious dialogue with the Jewish faith and Jewish communities—Tomasz Terlikowski, a journalist and commentator with a strong Twitter following (31,000 followers) who embraced the official Polish Episcopate announcement with the following tweet:

> *Dzień Judaizmu przypomina nam o naszych korzeniach, a może jeszcze lepiej powiedzieć o tym, kim naprawdę jesteśmy. Każdy chrześcijanin jest w istocie duchowo Żydem. Jeśli nim nie jest, to nie rozumie ani własnej wiary, ani nie akceptuje Jezusa. Warto o tym—w tym dniu—pamiętać* (https://twitter.com/tterlikowski/status/1350773421106364 417, accessed on 30 January 2021)

> [ENG: *The Day of Judaism reminds us about our own roots and can help us define who we really are. Every Christian is in fact, in his soul, a Jew. If one does not feel that way, one does not understand neither one's own faith or Jesus. On such day, it is worth remembering it.*]

The tweet above successfully managed to push social media noise aside (although it did not erase it completely). Apart from the immediate network of Terlikowski's supporters, an important second cluster emerged which engaged with Terlikowski and—by disagreeing with the author within a framework of civil conversation—initiated a debate with supporters of both sides voicing their affiliation through comments, retweets, and likes. This introduced additional layers and depth to the reflection on entangled histories of Judaism and Christianity. The tweet came from Robert Tekieli, a prominent conservative journalist. His affiliation with Church, as a lay supporter, is indisputable and reflected in his first message in which the official tweet from Polish Episcopate is mentioned and retweeted. However, this initial tweet is followed by another one which does not directly

link itself with Terlikowski via the mechanism of retweets and mentions, although it bears an apparent trace of direct response to the prominent tweet:

> *Więc, powtórzę: przy okazji Dnia judaizmu nie opowiadajmy bzdur. Nie jesteśmy duchowo ani żydami ani tym bardziej w jakimkolwiek innym sensie Żydami. Duchowo jesteśmy uczniami i braćmi Jezusa, którego żydzi odrzucili.*
>
> *Mam też pytanie, czy judaizm obchodzi Dzień katolicyzmu?* ([https://twitter.com/RobertTekieli/status/1350783351792295939](https://twitter.com/RobertTekieli/status/1350783351792295939), accessed on 30 January 2020)
>
> [ENG: *So let me repeat myself: on the occasion of the Day of Judaism let us not talk rubbish. We are neither spiritually nor in any other sense Jewish. Spiritually we are the disciples and brothers of Jesus, whom Jews rejected.*
>
> *I also have a question, does Judaism celebrate a Day of Catholicism?*]

Tekieli (13,000 followers) embraced the idea of the Day of Judaism, although questions bidirectionally of the inter-religious dialogue in a tone similar to open opponents of the idea from data gathered in 2020. A similar attitude was expressed by another prominent conservative account of Krzysztof Bosak (22,000 followers). However, because as prominent users these accounts link their voice with the official Episcopates announcement and indirectly refer to a very positive stance towards the Day of Judaism expressed by Terlikowski, the discursive map of 2021 looks much different than the year before. It is closer to a balanced conversation where different voices are expressed but neither of them is dominant. With communities represented by Terlikowski and Tekieli, much more vocal this time, the right wing, revisionists, neo-conservatist clusters were situated at less prominent positions, as seen in Figure 3.

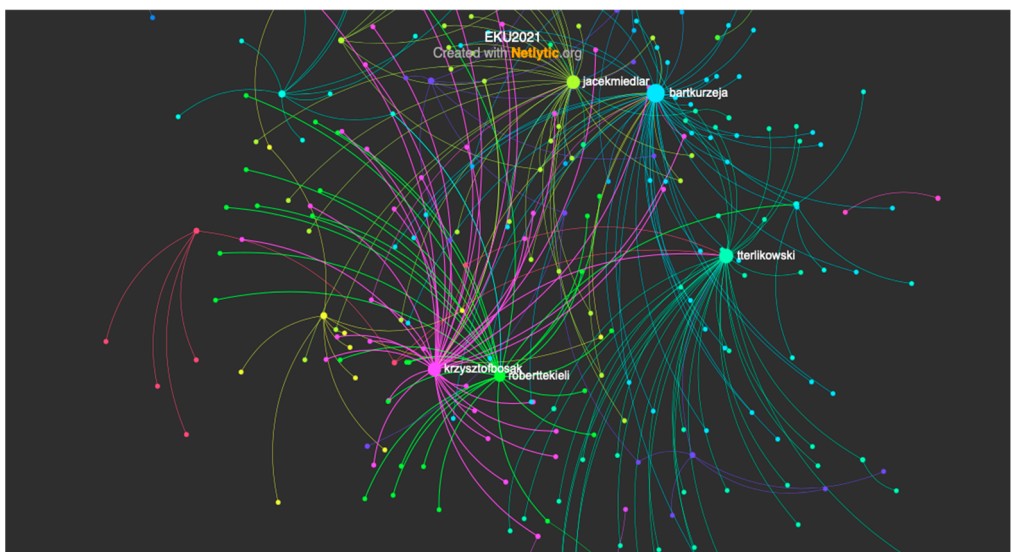

**Figure 3.** The discursive map of the Twitter conversation following gate announcements of the Days of Judaism by the Polish Episcopate. A strongly positive message (Terlikowski) and moderate conservative opponents of the message (Tekieli, Bosak) are situated in the center of the conversation. Anti-progressivist and right-wing clusters (entered around jacekmiedlar and bartkurzeja accounts) are pushed to the Northern edge of the map.

## 3. Results

While one must acknowledge the disruption of the unidirectional "one-to-many interface of mass media discourse" ([KhosraviNik 2017](#), p. 582), caused by the emergence of the many-to-many network model, one might also assume that—to some extent—the network is self-regulating. Far from being inclusive, the discussion we studied and mapped did not ignite extremist expressions. It becomes apparent that Twitter users who do engage in conversation around the Church's announcements are neither sworn enemies nor indifferent to the life of the Church in Poland. However, their engagement utilizes



affordances of social media, Twitter in particular, which allows for a strong preference of sentiment over content (Laniado and Mika 2010; González-Ibánez et al. 2011; Kouloumpis et al. 2011). Of course, and unfortunately, the sentiment level can reach an overdrive and users responsible for it can be perceived (by network analysis algorithms) as prominent players within the discourse. This, in turn, potentially skews the overall picture of the conversation. An anonymous user who scores a higher number of mentions and retweets can rise to prominence and effectively "hijack the hashtag" (Meraz 2017, p. 312) by diverting attention to itself. On the other hand, a tweet coming from a moderate or sympathetic user and intended to ignite a genuine exchange of arguments can be potentially embraced by representatives of much more radical views, and by retweets and mentions gravitate towards more radical clusters of the network.

A bad reputation of social media as a platform not fit for dialogue, balanced discussion, and constructive criticism can be easily confirmed when a controversial subject, or an issue with a potential to ignite cultural taboos, pre-conceptions, and false ideas is put at its center. Celebration of the Day of Judaism in the Polish Church is a telling example of such a subject. Nevertheless, the conclusions of our study are not entirely in line with the general understanding of how toxic and polarizing social media can be. A clear, balanced, personal statement that gives testimony to one's own embracement (Terlikowski) or doubts (Tekieli) about the subject in question is able to set a tone, provide an example, and initiate a dialog that is constructive and in the spirit of the inter-religious exchange. A blank polarity of official Church announcement and anonymous, negative feedback on the Twittersphere, built upon various agendas and metapragmatic motifs (Petykó 2018, p. 392), can be overcome. Once this happens, a stage for open discussion within the Catholic community emerges. Such a stage would be filled even more fully if Jewish online communities are also engaged in the conversation. This might happen one day because on social media, such engagement is always one click away.

One needs to ask if the Polish Twittersphere reactions on Episcopate's announcements of the Days of Judaism are anti-Semitic (Benz 2004, pp. 943–45; Neugröschel 2021, pp. 175–76). Contrary to the framing that social media opponents of the inter-religious dialog tend to give to their messages, implying that what they care about is "tradition" or "true Church", and what they do oppose is only modernization and political correctness, the answer to the question is—unfortunately—positive. The Episcopate's announcements of the Days of Islam, which fell within the same month of January, drew much less reaction on Twitter (65 tweets within two weeks after the initial posting) and proved that the prospect of inter-faith dialogue with Polish Muslims, or Muslims in general, did not cause as much controversy as in the case of the Jewish Faith.

**Author Contributions:** Conceptualization, M.P. and A.G.; Methodology, M.P.; Software, M.P.; Visualization, M.P.; Writing—original draft, M.P. and A.G.; Writing—review & editing, M.P. and A.G. Both authors have read and agreed to the published version of the manuscript.

**Funding:** This research received no external funding.

**Data Availability Statement:** The data used throughout this article that were rendered through the Twitter API within TADS tools are by license publicly available and accessible via the following url: https://hawksey.info/tagsexplorer/?key=1ByPc1j9WC_sC7btHjL29lysiHyApH3-w2LgmYmB9 3h8&gid=1552889159 (accessed on 30 January 2020 and 30 January 2021). Further visualization materials are available upon request.

**Conflicts of Interest:** The authors declare no conflict of interest.

## Notes

[1]   EpiskopatNews is a Twitter account founded in 2015 under the name @EpiskopatNews. The main purpose of this information stream is to promote in the media the teaching of the Polish Bishops' Conference and to inform about current events in the life of the Church in Poland and in the world. The account is run by the Press Office of the Polish Bishops' Conference, it is the most authoritative and influential source of information for Catholic and lay media, and is conducted in several languages.

²      The word "Rzydzi" (for Jews, correct spelling "Żydzi") is most likely deliberately misspelled here in a derogatory fashion in order to raise the expressiveness of this most toxic of more than 1000 analyzed tweets.

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
