# Peer review of "From Disruption to Dialog: Days of Judaism on Polish Twitter"

_religions, doi:10.3390/rel12100828_

Round 1

Reviewer 1 Report

The paper engages in a descriptive analysis of networks of “gatekeepers” and “gatewatchers” in a social media platform (Twitter) that exchange messages over the Polish Catholic Church’s annual “Day of Judaism”. Using Social Network Analysis and Discourse Analysis methods, the author/s identify key Twitter users, analyze their different discursive styles and dissect and map out their Twitter interactions. The paper’s conclusions assert that whereas the networks in question confirm Twitter as a social media tool where unbalanced and “toxic” discussions take place, they also provide examples of online communication strategies that may me conducive to inter-religious dialogue. Interesting and methodologically sound as it is, the paper however, does not come across as a good fit for the journal or the special issue (i.e. media-based participation in religious rituals before, during, and after the pandemic). More specifically:

-the paper offers a description of the Day of Judaism (ls. 186-198) but lacks a more substantial and contextual account of this event within the Polish Catholic Church, its historical background, its inter-religious profile, general and specific objectives, specific agents involved in its organization, history of clerical and lay participation, reception among the non-Catholic population, etc.

- the paper also lacks an account of this annual event’s relevance as a research subject vis-a-vis other events in/by the Catholic Church in Poland that very likely (or not?) are twitted on the internet

-there is engagement neither with the vast literature on inter-religious dialogue nor with the growing literature on inter-religious dialogue & online media.  

-there is no engagement with the literature on anti-semitism in Poland either

-the analysis identifies prominent agents with different religious-political profiles yet there is no substantial discussion on those profiles either.

-regarding the special issue’s focus, the paper does not really address the Covid pandemic as part of the context or the research subject.

As the paper’s analysis and conclusions tend to be dominated by the author/s’ engagement with the methodological literature, the paper could be a better fit for journals specialized in Social Network Analysis or new/online/digital media studies.

Author Response

-the paper offers a description of the Day of Judaism (ls. 186-198) but lacks a more substantial and contextual account of this event within the Polish Catholic Church, its historical background, its inter-religious profile, general and specific objectives, specific agents involved in its organization, history of clerical and lay participation, reception among the non-Catholic population, etc.

Thank you so much for your suggestions. The goal of the paper, and its main subject, is focused around the conversation of Church with its congregation, and specifically it reflects on modes of engagement in the conversation affordances of social media platforms such as Twitter. This is why the historical context we present reflects on history and the development of Church’s approach towards mass media: from rejection to embracement of the ever evolving mass communication platforms as a space for dialogue with the faithful: starting from 1814 and ending with Web 2.0. 

The commemorations of Days of Judaism function as a trigger of the conversation dynamics on Twitter that we analyse, and are not the focus of our study. 

However, we absolutely agree that more background should be given the the Days of Judaism and – as suggested by 3 reviewers – we expanded the article in this respect. 

- the paper also lacks an account of this annual event’s relevance as a research subject vis-a-vis other events in/by the Catholic Church in Poland that very likely (or not?) are twitted on the internet

Instead of expanding our study to include other Church events and their reception on social media within the same year or at random intervals, we chose to support our findings by reflecting on commemorations of the single even but over 2 consecutive years. We believe this approach can be more successful. By comparing the exchange between official Church announcements and the conversation it enacts over the course of 2 year we were able to discover a pattern of successful engagement and a recipe for a form of moderation of the conversation within a medium that is extremely hard to moderate (namely that active participation of individual supporters of the Church message can turn the potential toxicity of twittersphere into a constructive exchange of the faithful, even if they do not share the same ideas about the event).

-there is engagement neither with the vast literature on inter-religious dialogue nor with the growing literature on inter-religious dialogue & online media. 

As much as we would like to broaden the context of our study to embrace the general subject of inter-religious dialogue on the Web and costal media, this is not the primary focus of the study. The title of the article, “From Disruption to Dialog”, refers to the dialogue between the Church and the faithful. We only ask questions, if the optimal level of the conversation, carried “in the spirit” of inter-religious dialogue, expressed by ideas and intentions behind Days of Judaism, can be achieved. The answers we find relate to communication patterns and strategies we identify as possible and desirable. Inter-religious dialogue serves as an inspiration and a blue-print for the advocated constructive discussion on twittersphere. 

-there is no engagement with the literature on anti-semitism in Poland either

Unfortunately, anti-semitism is a constant ingredient of Internet trolling. The phenomena of “hashtag hijacking” and other forms of attention grabbing by social media users who redirect the discussion do often employ elements of racism and anti-semitism. Although some of the gather data bare traces of anti-semitism, such us derogatory spelling of “rzydzi” instead of “Żydzi” in one of the most toxic tweets, we found it to be an exception. However, we will include a couple of references to online anti-semitism in general in our bibliography. 

-the analysis identifies prominent agents with different religious-political profiles yet there is no substantial discussion on those profiles either

We have concluded that majority of the users engaged in the conversations that followed announcements of the Days of Judaism were Catholics, with differences between them distributed along the axis of conservatism and “progressivist” Catholicism on the one hand and the nationalist – internationalist axis, on the other. 

-regarding the special issue’s focus, the paper does not really address the Covid pandemic as part of the context or the research subject.

The second year of our data gathering was in the mids of the second wave of the covid pandemic and as such it produced a much higher user engagement with the announcement of the Days of Judaism. This quantitative change translated into a qualitative one: the discussion under pandemic was more healthy and more constructive. We have now added a mention about this in the analysis part. 

– As the paper’s analysis and conclusions tend to be dominated by the author/s’ engagement with the methodological literature, the paper could be a better fit for journals specialized in Social Network Analysis or new/online/digital media studies.

In our conclusions we also hint at possible improvement of the Church’s engagement within the twittersphere in order to improve the exchange between the institution and individuals, and this proposals are backed by examples we analyse in the network study part. These findings are relevant and might be of use to the Church’s media outlets, press rooms etc. This is why we would be honoured to present our findings in the journal of religions rather, than in social science or digital humanities journals. 

Reviewer 2 Report

The cognitively interesting question contained in the title: „From Disruption to Dialogue: Days of Judaism on Polish Twitter”, which set the current research problem. I appreciate the research effort. I fully embrace interdisciplinary research. I note sound and balanced judgments. I will highlight and expand on other comments in the following paragraphs.

  1. Introduction (22-91). The article provides a good, tested yet innovative starting point for the research: Twitter and other social media platforms are not only networking tools, but above all linguistic tools. Today when interdisciplinary research is creative: combining two methodological approaches is legitimate. The analysis of social networks, indeed focuses on the relations between users, their cumulative position in the discursive field, allowed to detect the ad hoc formation of online communities, applied to graph and network theory. Complementary is the second approach, which draws on linguistic research on new media, their rhetorical possibilities and limitations, to apply the classical notions of rhetorical effects of ethos, pathos and logos in the context of social media. For the reviewer, what is most original here is a study considered in the broader historical context of the Church's public conversation in contemporary media.

What is essential to complete?

  1. (184-185). Show more broadly what is Judaism Day? Its history, essence, and scope need to be briefly elaborated. It is an important event for Christian-Jewish dialogue, but perhaps more importantly, it deepens the knowledge of the Jewish religion by Christians themselves. This is the implementation of the Second Vatican Council. However, only a few countries are implementing it: Austria, Italy, the Netherlands and Poland, and perhaps some other. The author might add a linguistic analysis, in which he specializes, and consider the name: in German: „Tag des Judentums”; in Polish: „Dzien Judaizmu”; and in Italian, perhaps more clearly and creatively: „Giornata per l’approfondimento e lo sviluppo del dialogo tra cattolici ed ebrei”.
  2. (185) In one sentence add that the Day of Judaism precedes and tries to integrate itself with the great worldwide event of the Week of Prayer for Christian Unity, which has borne much important fruit for many religions and which has been in operation since 1908 thanks to Rev. P. Wattson.
  3. There is a readable description of figure 3 (441-445). It is illegible, there is a description of figure 2. (351-353) to be corrected. Either use U1, U2, U3 on the graphic, or in the description, along the lines of figure 1 in parentheses (czarna limuzyna, etc.).

What can be supplemented? (I leave to the author's discretion)

  1. (79). EpiskopatNews - a Twitter account founded in 2015 under the name @EpiskopatNews, the main purpose of this information stream is to promote in the media the teaching of the Polish Bishops' Conference and to inform about current events in the life of the Church in Poland and in the world. Run by the Press Office of the Polish Bishops' Conference, it is the most authoritative and influential source of information for Catholic and lay media, and is conducted in several languages.
  2. (95) I would delete the words "one-sided even" and add (96) in addition to oral communication, life testimonies also the term "letters", or "correspondence", because in the Roman Empire letters reached three continents in two weeks and we have not only letters in Scripture which are an example of the great communication of that time but also dialogue between communities of believers or individual leaders of religious communication.

7 When for several days there has been a conflict not only in the media between Israel and Poland (The Jerusalem Post@Jerusalem_Post 14.08: Foreign Minister @yairlapid recalls Israel's embassy head from Poland following the approval of the Holocaust anti-restitution bill.). This article is a brick that builds a path to dialogue. It can be an example of reliable science and may it also have a social impact.

Author Response

ANSWERS TO Reviewer 2

What is essential to complete?

  1. (184-185). Show more broadly what is Judaism Day? Its history, essence, and scope need to be briefly elaborated. It is an important event for Christian-Jewish dialogue, but perhaps more importantly, it deepens the knowledge of the Jewish religion by Christians themselves. This is the implementation of the Second Vatican Council. However, only a few countries are implementing it: Austria, Italy, the Netherlands and Poland, and perhaps some other. The author might add a linguistic analysis, in which he specializes, and consider the name: in German: „Tag des Judentums”; in Polish: „Dzien Judaizmu”; and in Italian, perhaps more clearly and creatively: „Giornata per l’approfondimento e lo sviluppo del dialogo tra cattolici ed ebrei”.

We have included additional parts to explain the backgrounds of the Days of Judaism

  1. (185) In one sentence add that the Day of Judaism precedes and tries to integrate itself with the great worldwide event of the Week of Prayer for Christian Unity, which has borne much important fruit for many religions and which has been in operation since 1908 thanks to Rev. P. Wattson.

We have added the suggested. 

  1. There is a readable description of figure 3 (441-445). It is illegible, there is a description of figure 2. (351-353) to be corrected. Either use U1, U2, U3 on the graphic, or in the description, along the lines of figure 1 in parentheses (czarna limuzyna, etc.).

Thank you so much for this suggestion. I have now labelled U1, U2, U3, U4 on illustrations marked as Figure1 and Figure 2. 

What can be supplemented 

(79). EpiskopatNews - a Twitter account founded in 2015 under the name @EpiskopatNews, the main purpose of this information stream is to promote in the media the teaching of the Polish Bishops' Conference and to inform about current events in the life of the Church in Poland and in the world. Run by the Press Office of the Polish Bishops' Conference, it is the most authoritative and influential source of information for Catholic and lay media, and is conducted in several languages.

Thank you very much. This information was added in the endnotes.

(95) I would delete the words "one-sided even" and add (96) in addition to oral communication, life testimonies also the term "letters", or "correspondence", because in the Roman Empire letters reached three continents in two weeks and we have not only letters in Scripture which are an example of the great communication of that time but also dialogue between communities of believers or individual leaders of religious communication.

Removed “one-sided” and added “letters” in the following sentence. 

Reviewer 3 Report

  1. I suggest splitting a very long Introduction and separate parts of Background and Literature Review.
  2. It is necessary to expand the section on the history of the Day of Judaism in the Catholic Church in Poland. It is an initiative with a long history, it is important to recall its chronicle. As an aside, it can be mentioned that the Catholic Church in Poland initiated a twin formula, called the Day of Islam in the Catholic Church in Poland.
  3. The author / authors should also supplement the literature with reports related to the history of the Day of Judaism.
  4. Line 326 - why is „Rzydzi” written? Was this the case in the original text?

Summing up, the article is a reliable summary of interesting research. It should be published after the changes mentioned are made. Another plus point is the fact that new literature was used in the work.

Author Response

ANSWERS To Reviewer 3 

The introduction is split now into two subsections 1.1 and 1.2 

  1. It is necessary to expand the section on the history of the Day of Judaism in the Catholic Church in Poland. It is an initiative with a long history, it is important to recall its chronicle. As an aside, it can be mentioned that the Catholic Church in Poland initiated a twin formula, called the Day of Islam in the Catholic Church in Poland.

We have expanded the section on the history of the Days of Judaism.

  1. The author / authors should also supplement the literature with reports related to the history of the Day of Judaism.

We have added related literature in the bibliography .

  1. Line 326 - why is „Rzydzi” written? Was this the case in the original text?

Thank you. Yes it is “Rzydzi”, I have added the “sic!” mark just behind this – deliberately – misspelled word. An endnote which explains derogatory fashion of the misspelling was added. 

Neugröschel, Marc. "Redemption Online: Antisemitism and Anti-Americanism in Social Media: ". Volume 5 Confronting Antisemitism in Modern Media, the Legal and Political Worlds, edited by Armin Lange, Kerstin Mayerhofer, Dina Porat and Lawrence H. Schiffman, Berlin, Boston: De Gruyter, 2021, pp. 175-200. https://doi.org/10.1515/9783110671964-012

          1. I suggest splitting a very long Introduction and separate parts of Background and Literature Review.

Lyn Heitkamp, Kristina. 2015. Confronting Anti-Semitism

Speak Up! Confronting Discrimination in Your Daily Life. 

Round 2

Reviewer 1 Report

The new sections 1.3 and 1.4 in the revised version amend some of the manuscript's major weaknesses. However, the content in the new section 1.3, could have included several works/authors that have studied inter-religious dialogues from social science perspectives. The paper might well be an interesting contribution for pastoral debates within the church, its appeal to a wider audience of social scientists (except for those specialized in digital media studies) still seems to me rather low though.